# Semi-crowdsourced Clustering with Deep Generative Models

**Yucen Luo, Tian Tian, Jiaxin Shi, Jun Zhu,\* Bo Zhang**
Dept. of Comp. Sci. & Tech., Institute for AI, THBI Lab, BNRist Center,
State Key Lab for Intell. Tech. & Sys., Tsinghua University, Beijing, China
{luoyc15,shijx15}@mails.tsinghua.edu.cn, rossowhite@163.com
{dcszj,dcszb}@mail.tsinghua.edu.cn

## Abstract

We consider the semi-supervised clustering problem where crowdsourcing provides noisy information about the pairwise comparisons on a small subset of data, i.e., whether a sample pair is in the same cluster. We propose a new approach that includes a deep generative model (DGM) to characterize low-level features of the data, and a statistical relational model for noisy pairwise annotations on its subset. The two parts share the latent variables. To make the model automatically trade-off between its complexity and fitting data, we also develop its fully Bayesian variant. The challenge of inference is addressed by fast (natural-gradient) stochastic variational inference algorithms, where we effectively combine variational message passing for the relational part and amortized learning of the DGM under a unified framework. Empirical results on synthetic and real-world datasets show that our model outperforms previous crowdsourced clustering methods.

## 1 Introduction

Clustering is a classic data analysis problem when the taxonomy of data is unknown in advance. Its main goal is to divide samples into disjunct clusters based on the similarity between them. Clustering is useful in various application areas including computer vision [21], bioinformatics [28], anomaly detection [2], etc. When the feature vectors of samples are observed, most clustering algorithms require a similarity or distance metric defined in the feature space, so that the optimization objective can be built. Since different metrics may result in entirely different clustering results, and general geometry metrics may not meet the intention of the tasks' designer, many clustering approaches learn the metric from the side-information provided by domain experts [30], thus the manual labeling procedure of experts could be a bottleneck for the learning pipeline.

Crowdsourcing is an efficient way to collect human feedbacks [12]. It distributes micro-tasks to a group of ordinal web workers in parallel, so the whole task can be done fast with relatively low cost. It has been used on annotating large-scale machine learning datasets such as ImageNet [6], and can also be used to collect side-information for clustering. However, directly collecting labels from crowds may lead to low-quality results due to the lack of expertise of workers. Consider an example of labeling a set of images of flowers from different species. One could show images to the web workers and ask them to identify the corresponding species, but such tasks require the workers to be experts in identifying the flowers and have all the species in their minds, which is not always possible. A more reasonable and easier task is to ask the workers to compare pairs of flower images and to answer whether they are in the same species or not. Then specific clustering methods are required to discover the clusters from the noisy feedbacks.

To solve above clustering problems with pairwise similarity labels between samples from the crowds, Crowdclustering [8] discovers the clusters within the dataset using a Bayesian hierarchical model.

---

By explicitly modeling the mistakes and preferences of web workers, the outputs will match the human consciousness of the clustering tasks. This method reduces the labeling cost to a great degree compared with expert labeling. However, the cost still grows quadratically as the dataset size grows, so it is still only suitable for small datasets. In this work, we move one step further and consider the *semi-supervised crowdclustering* problem that jointly models the feature vectors and the crowdsourced pairwise labels for only a subset of samples. When we control the size of the subset to be labeled by crowds, the total labeling budget and time can be controlled. A similar problem has been discussed by [31], while the authors use a linear similarity function defined on the low-level object features, and ignore the noise and inter-worker variations in the manual annotations.

Different from existing approaches, we propose a semi-supervised deep Bayesian model to jointly model the generation of the labels and the raw features for both crowd-labeled and unlabeled samples. Instead of the direct usage of low-level features, we build a flexible deep generative model (DGM) to capture the latent representation of data, which is more suitable to express the semantic similarity than the low-level features. The crowdsourced pairwise labels are modeled by a statistical relational model, and the two parts (i.e., DGM and the relational model) share the same latent variables. We also investigate the fully Bayesian variant of this model so that it can automatically control its complexity. Due to the intractability of exact inference, we develop fast (natural-gradient) stochastic variational inference algorithms. To address the challenges in fully Bayesian inference over model parameters, we effectively combine variational message passing and natural gradient updates for the conjugate part (i.e., the relational model and the mixture model) and amortized learning of the nonconjugate part (i.e., DGM) under a unified framework. Empirical results on synthetic and real-world datasets show that our model outperforms previous crowdsourced clustering methods.

## 2 Semi-crowdsourced deep clustering

In this section, we propose the semi-crowdsourced clustering with deep generative models for directly modeling the raw data, which enables end-to-end training. We call the model *Semi-crowdsourced Deep Clustering* (SCDC), whose graphical model is shown in Figure 1. This model is composed of two parts: the raw data model handles the generative process of the observations $\mathbf{O}$; the crowdsourcing behavior model on labels $\mathbf{L}$ describes the labeling procedure of the workers. The details for each part will be introduced below.

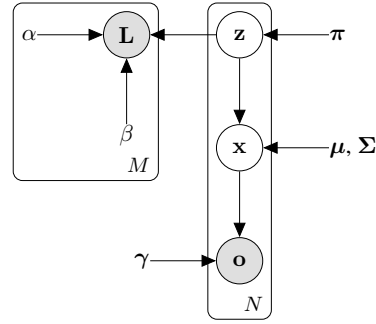

Figure 1: Semi-crowdsourced Deep Clustering (SCDC).

### 2.1 Model the raw data – deep generative models

We denote the raw data observations by $\mathbf{O} = \{\mathbf{o}_1, ..., \mathbf{o}_N\}$. For images, $\mathbf{o}_n \in \mathbb{R}^D$ denotes the pixel values. For each data point $\mathbf{o}_n$ we have a corresponding latent variable $\mathbf{x}_n \in \mathbb{R}^d$ and $p(\mathbf{o}_n|\mathbf{x}_n, \boldsymbol{\gamma})$ is a flexible neural network density model parametrized by $\boldsymbol{\gamma}$. $p(\mathbf{x}_n|\mathbf{z}_n; \boldsymbol{\mu}, \boldsymbol{\Sigma})$ is a Gaussian mixture where $\mathbf{z}_n$ comprises a 1-of-$K$ binary vector with elements $z_{nk}$ for $k = 1, ..., K$. Here $K$ denotes the number of clusters. We denote the local latent variables by $\mathbf{X} = \{\mathbf{x}_1, ..., \mathbf{x}_N\}$, $\mathbf{Z} = \{\mathbf{z}_1, ..., \mathbf{z}_N\}$. When real-valued observations are given, the generative process is as follows:

$$p(\mathbf{Z}; \boldsymbol{\pi}) = \prod_{n=1}^{N} p(\mathbf{z}_n; \boldsymbol{\pi}) = \prod_{n=1}^{N} \prod_{k=1}^{K} \pi_k^{z_{nk}}, \quad p(\mathbf{X}|\mathbf{Z}; \boldsymbol{\mu}, \boldsymbol{\Sigma}) = \prod_{n=1}^{N} \prod_{k=1}^{K} \mathcal{N}(\mathbf{x}_n; \boldsymbol{\mu}_k, \boldsymbol{\Sigma}_k)^{z_{nk}},$$

$$p(\mathbf{O}|\mathbf{X}; \boldsymbol{\gamma}) = \prod_{n=1}^{N} \mathcal{N}(\mathbf{o}_n|\boldsymbol{\mu}_{\boldsymbol{\gamma}}(\mathbf{x}_n), \mathrm{diag}(\boldsymbol{\sigma}_{\boldsymbol{\gamma}}^2(\mathbf{x}_n))),$$

where $\boldsymbol{\mu}_{\boldsymbol{\gamma}}(\cdot)$ and $\boldsymbol{\sigma}_{\boldsymbol{\gamma}}^2(\cdot)$ are two neural networks parameterized by $\boldsymbol{\gamma}$. For other types of observations $\mathbf{O}$, $p(\mathbf{o}|\mathbf{x}; \boldsymbol{\gamma})$ can be other distributions, e.g. Bernoulli distribution for binary observations. In general, our model is a deep generative model with structured latent variables.

### 2.2 Model the behavior of each worker – two-coin Dawid-Skene model

We collect pairwise annotations provided by $M$ workers. A partially observed $\mathbf{L}^{(m)} \in \{0, 1, \texttt{NULL}\}^{N_l \times N_l}$ is the annotation matrix of the $m$-th worker, where $N_l$ is the number of an-

notated data points. For observation pairs $(\mathbf{o}_i, \mathbf{o}_j), i \neq j, L_{ij}^{(m)} = 1$ represents that the $m$-th worker provides a must-link (ML) constraint, which means observations $i$ and $j$ belong to a same cluster, $L_{ij}^{(m)} = 0$ represents cannot-link (CL) constraint, which means observations $i$ and $j$ belong to different clusters, and NULL represents that $L_{ij}^{(m)}$ is not observed. It is obvious that $\mathbf{L}^{(m)}$ is symmetric, i.e., $L_{ij}^{(m)} = L_{ji}^{(m)}, \forall i, j, m$. Self-edges are not allowed, i.e., $L_{ii}^{(m)} = \text{NULL}, \forall i$.

Among all the $N$ data observations $\mathbf{O}$, we only crowdsource pairwise annotations for a small portion of $\mathbf{O}$, denoted by $\mathbf{O}_L$. Each worker only provides annotations to a small amount of items in $\mathbf{O}_L$ and the annotation accuracies of non-expert workers may vary with observations and levels of expertise. We adopt the two-coin Dawid-Skene model for annotators from [18] and develop a probabilistic model by explicitly modeling the uncertainty of each worker. Specifically, the uncertainty of the $m$-th worker can be characterized by accuracy parameters $(\alpha_m, \beta_m)$, where $\alpha_m$ represents sensitivity, which means the probability of providing ML constraints for sample pairs belonging to the same cluster. And $\beta_m$ is the $m$-th worker's specificity, which means the probability of providing CL constraints for sample pairs from different clusters. Let $\boldsymbol{\alpha} = \{\alpha_1, ..., \alpha_M\}$ and $\boldsymbol{\beta} = \{\beta_1, ..., \beta_M\}$. The likelihood is defined as

$$p(L_{ij}^{(m)}|\mathbf{z}_i, \mathbf{z}_j; \alpha_m, \beta_m) = \text{Bern}(L_{ij}^{(m)}|\alpha_m)^{\mathbf{z}_i^\top \mathbf{z}_j} \text{Bern}(L_{ij}^{(m)}|1-\beta_m)^{1-\mathbf{z}_i^\top \mathbf{z}_j}, \tag{1}$$

or equivalently, $p(L_{ij}^{(m)} = 1|\mathbf{z}_i = \mathbf{z}_j, \alpha_m) = \alpha_m, p(L_{ij}^{(m)} = 0|\mathbf{z}_i \neq \mathbf{z}_j, \beta_m) = \beta_m$. To simplify the notation, we define $I_{ij}^{(m)} = \mathbb{I}[L_{ij}^{(m)} \neq \text{NULL}]$. Using the symmetry of $\mathbf{L}^{(m)}$, the total likelihood of annotations can be written

$$p(\mathbf{L}|\mathbf{Z}; \boldsymbol{\alpha}, \boldsymbol{\beta}) = \prod_{m=1}^{M} \prod_{1 \leq i < j \leq N} p(L_{ij}^{(m)}|\mathbf{z}_i, \mathbf{z}_j; \alpha_m, \beta_m)^{I_{ij}^{(m)}}. \tag{2}$$

## 2.3 Amortized variational inference

As described above, the parameters in the semi-crowdsourced deep clustering model include $\boldsymbol{\pi} \in [0,1]^K, \boldsymbol{\mu} \in \mathbb{R}^{K \times d}, \boldsymbol{\Sigma} \in \mathbb{R}^{K \times d \times d}, \boldsymbol{\alpha}, \boldsymbol{\beta} \in [0,1]^M$, and the parameters of neural networks $\boldsymbol{\gamma}$. Let $\boldsymbol{\Theta} = \{\boldsymbol{\pi}, \boldsymbol{\mu}, \boldsymbol{\Sigma}, \boldsymbol{\alpha}, \boldsymbol{\beta}\}$, the overall joint likelihood of the model is

$$p(\mathbf{Z}, \mathbf{X}, \mathbf{O}, \mathbf{L}; \boldsymbol{\Theta}, \boldsymbol{\gamma}) = p(\mathbf{Z}; \boldsymbol{\pi})p(\mathbf{X}|\mathbf{Z}; \boldsymbol{\mu}, \boldsymbol{\Sigma})p(\mathbf{O}|\mathbf{X}; \boldsymbol{\gamma})p(\mathbf{L}|\mathbf{Z}; \boldsymbol{\alpha}, \boldsymbol{\beta}). \tag{3}$$

For this model, the learning objective is to maximize the variational lower bound $\mathcal{L}(\mathbf{O}, \mathbf{L})$ of the marginal log likelihood of the entire dataset $\log p(\mathbf{O}, \mathbf{L})$:

$$\log p(\mathbf{O}, \mathbf{L}) \geq \mathbb{E}_{q(\mathbf{Z}, \mathbf{X}|\mathbf{O})} [\log p(\mathbf{Z}, \mathbf{X}, \mathbf{O}, \mathbf{L}; \boldsymbol{\Theta}, \boldsymbol{\gamma}) - \log q(\mathbf{Z}, \mathbf{X}|\mathbf{O})] = \mathcal{L}(\mathbf{O}, \mathbf{L}; \boldsymbol{\Theta}, \boldsymbol{\gamma}, \boldsymbol{\phi}) \tag{4}$$

To deal with the non-conjugate likelihood $p(\mathbf{O}|\mathbf{X}; \boldsymbol{\gamma})$, we introduce inference networks for each of the latent variables $\mathbf{z}_n$ and $\mathbf{x}_n$. The inference networks are assumed to have a factorized form $q(\mathbf{z}_n, \mathbf{x}_n|\mathbf{o}_n) = q(\mathbf{z}_n|\mathbf{o}_n; \boldsymbol{\phi})q(\mathbf{x}_n|\mathbf{z}_n, \mathbf{o}_n; \boldsymbol{\phi})$, which are Categorical and Normal distributions, respectively:

$$q(\mathbf{z}_n|\mathbf{o}_n; \boldsymbol{\phi}) = \text{Cat}(\mathbf{z}_n; \boldsymbol{\pi}(\mathbf{o}_n; \boldsymbol{\phi})), q(\mathbf{x}_n|\mathbf{z}_n, \mathbf{o}_n; \boldsymbol{\phi}) = \mathcal{N}(\boldsymbol{\mu}(\mathbf{z}_n, \mathbf{o}_n; \boldsymbol{\phi}), \text{diag}(\boldsymbol{\sigma}^2(\mathbf{z}_n, \mathbf{o}_n; \boldsymbol{\phi}))),$$

where $\boldsymbol{\sigma}(\mathbf{z}_n, \mathbf{o}_n; \boldsymbol{\phi})$ is a vector of standard deviations and $\boldsymbol{\phi}$ denotes the inference networks parameters. Similar to the approach in [14], we can analytically sum over the discrete variables $\mathbf{z}_n$ in the lower bound and use the reparameterization trick to compute gradients w.r.t. to $\boldsymbol{\Theta}, \boldsymbol{\gamma}$ and $\boldsymbol{\phi}$.

The above objective sums over all data and annotations. For large datasets, we can conveniently use a stochastic version by approximating the lower bound with subsampled minibatches of data. Specifically, the variational lower bound is decomposed into two terms: $\mathcal{L}(\mathbf{L}, \mathbf{O}; \boldsymbol{\Theta}, \boldsymbol{\gamma}, \boldsymbol{\phi}) = \mathcal{L}_{\text{local}} + \mathcal{L}_{\text{rel}}$, where $\mathcal{L}_{\text{local}} = \sum_{n=1}^{N} \mathbb{E}_{q(\mathbf{z}_n, \mathbf{x}_n|\mathbf{o}_n)} [\log p(\mathbf{z}_n) + \log p(\mathbf{x}_n|\mathbf{z}_n) + \log p(\mathbf{o}_n|\mathbf{x}_n) - \log q(\mathbf{z}_n, \mathbf{x}_n|\mathbf{o}_n)]$, and $\mathcal{L}_{\text{rel}} = \sum_{m=1}^{M} \sum_{1 \leq i < j \leq N} I_{ij}^{(m)} \mathbb{E}_{q(\mathbf{z}_i, \mathbf{z}_j|\mathbf{o}_i, \mathbf{o}_j; \boldsymbol{\phi})} \log p(L_{ij}^{(m)}|\mathbf{z}_i, \mathbf{z}_j; \alpha_m, \beta_m)$. It is easy to derive an unbiased stochastic approximation of $\mathcal{L}_{\text{local}}$:

$$\mathcal{L}_{\text{local}} \approx \frac{N}{|B|} \sum_{n \in B} \mathbb{E}_{q(\mathbf{z}_n, \mathbf{x}_n|\mathbf{o}_n)} [\log p(\mathbf{z}_n) + \log p(\mathbf{x}_n|\mathbf{z}_n) + \log p(\mathbf{o}_n|\mathbf{x}_n) - \log q(\mathbf{z}_n, \mathbf{x}_n|\mathbf{o}_n)],$$

where $B$ is the sampled minibatch. For $\mathcal{L}_{\text{rel}}$, we can similarly randomly sample a minibatch $S$ of annotations: $\mathcal{L}_{\text{rel}} \approx \frac{N_a}{|S|} \sum_{(i,j,m) \in S} \mathbb{E}_{q(\mathbf{z}_i, \mathbf{z}_j|\mathbf{o}_i, \mathbf{o}_j; \boldsymbol{\phi})} \log p(L_{ij}^{(m)}|\mathbf{z}_i, \mathbf{z}_j; \alpha_m, \beta_m)$, where $N_a = \sum_{m=1}^{M} \sum_{1 \leq i < j \leq N} I_{ij}^{(m)}$ denotes the total number of annotations.

# 3 Natural gradient inference for the fully Bayesian model

In the previous section, the global parameters $\boldsymbol{\Theta} = \{\boldsymbol{\alpha}, \boldsymbol{\beta}, \boldsymbol{\pi}, \boldsymbol{\mu}, \boldsymbol{\Sigma}\}$ are assumed to be deterministic and are directly optimized by gradient descent. In this section, we propose a fully Bayesian variant of our model (BayesSCDC), which has an automatic trade-off between its complexity and fitting the data. There is no overfitting if we choose a large number $K$ of components in the mixture, in which case the variational treatment below can automatically determine the optimal number of mixture components. We develop fast natural-gradient stochastic variational inference algorithms for BayesSCDC, which effectively combines variational message passing for the conjugate structures (i.e., the relational part and the mixture part) and amortized learning of deep components (i.e., the deep generative model).

## 3.1 Fully Bayesian semi-crowdsourced deep clustering (BayesSCDC)

For the mixture model, we choose a Dirichlet prior over the mixing coefficients $\boldsymbol{\pi}$ and an independent Normal-Inverse-Wishart prior governing the mean and covariance $(\boldsymbol{\mu}, \boldsymbol{\Sigma})$ of each Gaussian component, given by

$$p(\boldsymbol{\pi}) = \text{Dir}(\boldsymbol{\pi}|\alpha_0) = C(\alpha_0) \prod_{k=1}^{K} \pi_k^{\alpha_0 - 1}, \quad p(\boldsymbol{\mu}, \boldsymbol{\Sigma}) = \prod_{k=1}^{K} \text{NIW}(\boldsymbol{\mu}_k, \boldsymbol{\Sigma}_k | \mathbf{m}, \kappa, \mathbf{S}, \nu), \quad (5)$$

where $\mathbf{m} \in \mathbb{R}^d$ is the location parameter, $\kappa > 0$ is the concentration, $\mathbf{S} \in \mathbb{R}^{d \times d}$ is the scale matrix (positive definite), and $\nu > d - 1$ is the degrees of freedom. The densities of $\boldsymbol{\pi}, \mathbf{z}, (\boldsymbol{\mu}, \boldsymbol{\Sigma}), \mathbf{x}$ can be written in the standard form of exponential families as:

$$p(\boldsymbol{\pi}) = \exp\left\{\langle \boldsymbol{\eta}_{\boldsymbol{\pi}}^0, \mathbf{t}(\boldsymbol{\pi}) \rangle - \log Z(\boldsymbol{\eta}_{\boldsymbol{\pi}}^0)\right\}, \quad p(\boldsymbol{\mu}, \boldsymbol{\Sigma}) = \exp\left\{\langle \boldsymbol{\eta}_{\boldsymbol{\mu}, \boldsymbol{\Sigma}}^0, \mathbf{t}(\boldsymbol{\mu}, \boldsymbol{\Sigma}) \rangle - \log Z(\boldsymbol{\eta}_{\boldsymbol{\mu}, \boldsymbol{\Sigma}}^0)\right\},$$

$$p(\mathbf{z}|\boldsymbol{\pi}) = \exp\left\{\langle \boldsymbol{\eta}_{\mathbf{z}}^0(\boldsymbol{\pi}), \mathbf{t}(\mathbf{z}) \rangle - \log Z(\boldsymbol{\eta}_{\mathbf{z}}^0(\boldsymbol{\pi}))\right\} = \exp\left\{\langle \mathbf{t}(\boldsymbol{\pi}), (\mathbf{t}(\mathbf{z}), 1) \rangle\right\},$$

$$p(\mathbf{x}|\mathbf{z}, \boldsymbol{\mu}, \boldsymbol{\Sigma}) = \exp\left\{\langle \mathbf{t}(\mathbf{z}), \mathbf{t}(\boldsymbol{\mu}, \boldsymbol{\Sigma})^\top (\mathbf{t}(\mathbf{x}), 1) \rangle\right\},$$

where $\boldsymbol{\eta}$ denotes the natural parameters, $\mathbf{t}(\cdot)$ denotes the sufficient statistics[2], and $\log Z(\cdot)$ denotes the log partition function.

For the relational model, we assume the accuracy parameters of all workers $(\boldsymbol{\alpha}, \boldsymbol{\beta})$ are drawn independently from common priors. We choose conjugate Beta priors for them as

$$p(\boldsymbol{\alpha}) = \prod_{m=1}^{M} p(\alpha_m) = \prod_{m=1}^{M} \text{Beta}(\tau_{\alpha_0^1}, \tau_{\alpha_0^2}), \quad p(\boldsymbol{\beta}) = \prod_{m=1}^{M} p(\beta_m) = \prod_{m=1}^{M} \text{Beta}(\tau_{\beta_0^1}, \tau_{\beta_0^2}). \quad (6)$$

We write the exponential family form of $p(\alpha_m)$ as: $p(\alpha_m) = \exp\left\{\langle \boldsymbol{\eta}_{\alpha_m}^0, \mathbf{t}(\alpha_m) \rangle - \log Z(\boldsymbol{\eta}_{\alpha_m}^0)\right\}$ ($p(\beta_m)$ is similar), where $\boldsymbol{\eta}_{\alpha_m}^0 = [\tau_{\alpha_0^1} - 1, \tau_{\alpha_0^2} - 1]^\top$ and $\mathbf{t}(\alpha_m) = [\log \alpha_m, \log(1 - \alpha_m)]^\top$.

## 3.2 Natural-gradient stochastic variational inference

The overall joint distribution of all of the hidden and observed variables takes the form:

$$p(\mathbf{L}^{(1:M)}, \mathbf{O}, \mathbf{X}, \mathbf{Z}, \boldsymbol{\Theta}; \boldsymbol{\gamma}) = p(\boldsymbol{\pi})p(\mathbf{Z}|\boldsymbol{\pi})p(\boldsymbol{\mu}, \boldsymbol{\Sigma})p(\mathbf{X}|\mathbf{Z}, \boldsymbol{\mu}, \boldsymbol{\Sigma})p(\mathbf{O}|\mathbf{X}; \boldsymbol{\gamma}) \quad (7)$$
$$\cdot p(\boldsymbol{\alpha})p(\boldsymbol{\beta})p(\mathbf{L}^{(1:M)}|\mathbf{Z}, \boldsymbol{\alpha}, \boldsymbol{\beta}).$$

Our learning objective is to maximize the marginal likelihood of observed data and pairwise annotations $\log p(\mathbf{O}, \mathbf{L}^{(1:M)})$. Exact posterior inference for this model is intractable. Thus we consider a mean-field variational family $q(\boldsymbol{\Theta}, \mathbf{Z}, \mathbf{X}) = q(\boldsymbol{\alpha})q(\boldsymbol{\beta})q(\mathbf{Z})q(\mathbf{X})q(\boldsymbol{\pi})q(\boldsymbol{\mu}, \boldsymbol{\Sigma})$. To simplify the notations, we write each variational distribution in its exponential family form: $q(\boldsymbol{\theta}) = \exp\left\{\langle \boldsymbol{\eta}_{\boldsymbol{\theta}}, \mathbf{t}(\boldsymbol{\theta}) \rangle - \log Z(\boldsymbol{\eta}_{\boldsymbol{\theta}})\right\}, \ \boldsymbol{\theta} \in \boldsymbol{\Theta} \cup \mathbf{Z} \cup \mathbf{X}$. The evidence lower bound (ELBO) $\mathcal{L}(\boldsymbol{\eta}_{\boldsymbol{\Theta}}, \boldsymbol{\eta}_{\mathbf{Z}}, \boldsymbol{\eta}_{\mathbf{X}}; \boldsymbol{\gamma})$ of $\log p(\mathbf{O}, \mathbf{L}^{(1:M)})$ is

$$\log p(\mathbf{O}, \mathbf{L}^{(1:M)}) \geq \mathcal{L}(\boldsymbol{\eta}_{\boldsymbol{\Theta}}, \boldsymbol{\eta}_{\mathbf{Z}}, \boldsymbol{\eta}_{\mathbf{X}}; \boldsymbol{\gamma}) \triangleq \mathbb{E}_{q(\boldsymbol{\Theta}, \mathbf{z}, \mathbf{x})} \log\left[\frac{p(\mathbf{L}^{(1:M)}, \mathbf{O}, \mathbf{X}, \mathbf{Z}, \boldsymbol{\Theta}; \boldsymbol{\gamma})}{q(\boldsymbol{\Theta})q(\mathbf{Z})q(\mathbf{X})}\right]. \quad (8)$$

In traditional mean-field variational inference for conjugate models, the optimal solution of maximizing eq. (8) over each variational parameter can be derived analytically given other parameters fixed, thus a coordinate ascent can be applied as an efficient message passing algorithm [27, 11]. However, it is not directly applicable to our model due to the non-conjugate observation likelihood $p(\mathbf{O}|\mathbf{X}; \boldsymbol{\gamma})$. Inspired by [13], we handle the non-conjugate likelihood by introducing recognition networks $r(\mathbf{o}_i; \boldsymbol{\phi})$. Different from SCDC in Section 2.3, the recognition networks here are used to form conjugate graphical model potentials:

$$\psi(\mathbf{x}_i; \mathbf{o}_i, \boldsymbol{\phi}) \triangleq \langle r(\mathbf{o}_i; \boldsymbol{\phi}), \mathbf{t}(\mathbf{x}_i) \rangle. \tag{9}$$

By replacing the non-conjugate likelihood $p(\mathbf{O}|\mathbf{X}; \boldsymbol{\gamma})$ in the original ELBO with a conjugate term defined by $\psi(\mathbf{x}_i; \mathbf{o}_i, \boldsymbol{\phi})$, we have the following surrogate objective $\widehat{\mathcal{L}}$:

$$\widehat{\mathcal{L}}(\boldsymbol{\eta}_{\boldsymbol{\Theta}}, \boldsymbol{\eta}_{\mathbf{Z}}, \boldsymbol{\eta}_{\mathbf{X}}; \boldsymbol{\phi}) \triangleq \mathbb{E}_{q(\boldsymbol{\Theta}, \mathbf{Z}, \mathbf{X})} \log \left[ \frac{p(\mathbf{L}^{(1:M)}, \mathbf{X}, \mathbf{Z}, \boldsymbol{\Theta}) \exp\{\psi(\mathbf{X}; \mathbf{O}, \boldsymbol{\phi})\}}{q(\boldsymbol{\Theta}) q(\mathbf{Z}) q(\mathbf{X})} \right]. \tag{10}$$

As we shall see, the surrogate objective $\widehat{\mathcal{L}}$ helps us exploit the conjugate structure in the model, thus enables a fast message-passing algorithm for these parts. Specifically, we can view eq. (10) as the ELBO of a conjugate graphical model with the same structure as in Fig. 1 (up to a constant). Similar to coordinate-ascent mean-field variational inference [11], we can derive the local partial optimizers of individual variational parameters as below.

The optimal solution for $q^*(\mathbf{X})$ factorizes over $n$, i.e., $q^*(\mathbf{X}) = \prod_{i=1}^{N} q^*(\mathbf{x}_i)$, and $q^*(\mathbf{x}_i)$ depends on the expected sufficient statistics of $(\boldsymbol{\mu}, \boldsymbol{\Sigma})$ and $\mathbf{z}_n$:

$$\log q^*(\mathbf{x}_i) = \mathbb{E}_{q(\boldsymbol{\mu}, \boldsymbol{\Sigma}) q(\mathbf{z}_i)} \log p(\mathbf{x}_i | \mathbf{z}_i, \boldsymbol{\mu}, \boldsymbol{\Sigma}) + \langle r(\mathbf{o}_i; \boldsymbol{\phi}), \mathbf{t}(\mathbf{x}_i) \rangle + \text{const}, \tag{11}$$

$$\boldsymbol{\eta}_{\mathbf{x}_i}^* = \mathbb{E}_{q(\boldsymbol{\mu}, \boldsymbol{\Sigma})} [\boldsymbol{\eta}_{\mathbf{x}_i}^0(\boldsymbol{\mu}, \boldsymbol{\Sigma})]^\top \mathbb{E}_{q(\mathbf{z}_i)} [\mathbf{t}(\mathbf{z}_i)] + r(\mathbf{o}_i; \boldsymbol{\phi}). \tag{12}$$

By further assuming a mean-field structure over $\mathbf{Z}$: $q^*(\mathbf{Z}) = \prod_{i=1}^{N} q^*(\mathbf{z}_i)$, we have the local partial optimizer for each single $q(\mathbf{z}_i)$ as

$$\log q^*(\mathbf{z}_i) = \mathbb{E}_{q(\boldsymbol{\pi})} \log p(\mathbf{z}_i | \boldsymbol{\pi}) + \mathbb{E}_{q(\boldsymbol{\mu}, \boldsymbol{\Sigma}) q(\mathbf{x}_i)} \log p(\mathbf{x}_i | \mathbf{z}_i, \boldsymbol{\mu}, \boldsymbol{\Sigma})$$
$$+ \mathbb{E}_{q(\boldsymbol{\alpha}) q(\boldsymbol{\beta}) q(\mathbf{Z}_{-i})} \left[ \log p(\mathbf{L}^{(1:M)} | \mathbf{Z}, \boldsymbol{\alpha}, \boldsymbol{\beta}) \right] + \text{const}, \tag{13}$$

$$\boldsymbol{\eta}_{\mathbf{z}_i}^* = \mathbb{E}_{q(\boldsymbol{\pi})} \mathbf{t}(\boldsymbol{\pi}) + \mathbb{E}_{q(\boldsymbol{\mu}, \boldsymbol{\Sigma})} [\mathbf{t}(\boldsymbol{\mu}, \boldsymbol{\Sigma})]^\top \mathbb{E}_{q(\mathbf{x}_i)} [(\mathbf{t}(\mathbf{x}_i), \mathbf{1})] + \sum_{m=1}^{M} \sum_{j=1}^{N} w_{ij}^{(m)} \mathbb{E}_{q(\mathbf{z}_j)} [\mathbf{t}(\mathbf{z}_j)], \tag{14}$$

where $w_{ij}^{(m)} = I_{ij}^{(m)} \mathbb{E}_{q(\boldsymbol{\alpha}, \boldsymbol{\beta})} \left[ \ln \frac{1-\alpha_m}{\beta_m} + L_{ij}^{(m)} \left( \ln \frac{\alpha_m}{1-\alpha_m} + \ln \frac{\beta_m}{1-\beta_m} \right) \right]$ is the weight of the message from $\mathbf{z}_j$ to $\mathbf{z}_i$. Using a block coordinate ascent algorithm that applies eqs. (12) and (14) alternatively, we can find the joint local partial optimizers $(\boldsymbol{\eta}_{\mathbf{Z}}^*(\boldsymbol{\eta}_{\boldsymbol{\Theta}}, \boldsymbol{\phi}), \boldsymbol{\eta}_{\mathbf{X}}^*(\boldsymbol{\eta}_{\boldsymbol{\Theta}}, \boldsymbol{\phi}))$ of $\widehat{\mathcal{L}}$ w.r.t. $(\boldsymbol{\eta}_{\mathbf{X}}, \boldsymbol{\eta}_{\mathbf{Z}})$ given other parameters fixed, i.e.,

$$\nabla_{\boldsymbol{\eta}_{\mathbf{Z}}} \widehat{\mathcal{L}}(\boldsymbol{\eta}_{\boldsymbol{\Theta}}, \boldsymbol{\eta}_{\mathbf{Z}}^*(\boldsymbol{\eta}_{\boldsymbol{\Theta}}, \boldsymbol{\phi}), \boldsymbol{\eta}_{\mathbf{X}}^*(\boldsymbol{\eta}_{\boldsymbol{\Theta}}, \boldsymbol{\phi}), \boldsymbol{\phi}) = 0, \quad \nabla_{\boldsymbol{\eta}_{\mathbf{X}}} \widehat{\mathcal{L}}(\boldsymbol{\eta}_{\boldsymbol{\Theta}}, \boldsymbol{\eta}_{\mathbf{Z}}^*(\boldsymbol{\eta}_{\boldsymbol{\Theta}}, \boldsymbol{\phi}), \boldsymbol{\eta}_{\mathbf{X}}^*(\boldsymbol{\eta}_{\boldsymbol{\Theta}}, \boldsymbol{\phi}), \boldsymbol{\phi}) = 0. \tag{15}$$

Plugging $(\boldsymbol{\eta}_{\mathbf{Z}}^*(\boldsymbol{\eta}_{\boldsymbol{\Theta}}, \boldsymbol{\phi}), \boldsymbol{\eta}_{\mathbf{X}}^*(\boldsymbol{\eta}_{\boldsymbol{\Theta}}, \boldsymbol{\phi}))$ back into $\mathcal{L}$, we define the final objective

$$\mathcal{J}(\boldsymbol{\eta}_{\boldsymbol{\Theta}}; \boldsymbol{\phi}, \boldsymbol{\gamma}) \triangleq \mathcal{L}(\boldsymbol{\eta}_{\boldsymbol{\Theta}}, \boldsymbol{\eta}_{\mathbf{Z}}^*(\boldsymbol{\eta}_{\boldsymbol{\Theta}}, \boldsymbol{\phi}), \boldsymbol{\eta}_{\mathbf{X}}^*(\boldsymbol{\eta}_{\boldsymbol{\Theta}}, \boldsymbol{\phi}), \boldsymbol{\gamma}). \tag{16}$$

As shown in [13], $\mathcal{J}(\boldsymbol{\eta}_{\boldsymbol{\Theta}}; \boldsymbol{\phi}, \boldsymbol{\gamma})$ lower-bounds the partially-optimized mean field objective, i.e., $\max_{\boldsymbol{\eta}_{\mathbf{X}}, \boldsymbol{\eta}_{\mathbf{Z}}} \mathcal{L}(\boldsymbol{\eta}_{\boldsymbol{\Theta}}, \boldsymbol{\eta}_{\mathbf{Z}}, \boldsymbol{\eta}_{\mathbf{X}}, \boldsymbol{\gamma}) \geq \mathcal{J}(\boldsymbol{\eta}_{\boldsymbol{\Theta}}, \boldsymbol{\gamma}, \boldsymbol{\phi})$, thus can serve as a variational objective itself. We compute the natural gradients of $\mathcal{J}$ w.r.t. the global variational parameters $\boldsymbol{\eta}_{\boldsymbol{\Theta}}$:

$$\widetilde{\nabla}_{\boldsymbol{\eta}_{\boldsymbol{\Theta}}} \mathcal{J} = \left[ \boldsymbol{\eta}_{\boldsymbol{\Theta}}^0 + \mathbb{E}_{q^*(\mathbf{Z}) q^*(\mathbf{X})} \left( \mathbf{t}(\mathbf{Z}, \mathbf{X}, \mathbf{L}^{(1:M)}), \mathbf{1} \right) - \boldsymbol{\eta}_{\boldsymbol{\Theta}} \right]$$
$$+ \left( \nabla_{\boldsymbol{\eta}_{\mathbf{Z}}, \boldsymbol{\eta}_{\mathbf{X}}} \mathcal{L}(\boldsymbol{\eta}_{\boldsymbol{\Theta}}, \boldsymbol{\eta}_{\mathbf{Z}}^*(\boldsymbol{\eta}_{\boldsymbol{\Theta}}, \boldsymbol{\phi}), \boldsymbol{\eta}_{\mathbf{X}}^*(\boldsymbol{\eta}_{\boldsymbol{\Theta}}, \boldsymbol{\phi}); \boldsymbol{\gamma}), \mathbf{0} \right). \tag{17}$$

Note that the first term in eq. (17) is the same as the formula of natural gradient in SVI [11], which is easy to compute, and the second term originates from the dependence of $\boldsymbol{\eta}_{\mathbf{Z}}^*, \boldsymbol{\eta}_{\mathbf{X}}^*$ on $\boldsymbol{\eta}_{\boldsymbol{\Theta}}$ and can be computed using the reparameterization trick. For other parameters $\boldsymbol{\phi}, \boldsymbol{\gamma}$, we can also get the gradients $\nabla_{\boldsymbol{\phi}} \mathcal{J}(\boldsymbol{\eta}_{\boldsymbol{\Theta}}; \boldsymbol{\phi}, \boldsymbol{\gamma})$ and $\nabla_{\boldsymbol{\gamma}} \mathcal{J}(\boldsymbol{\eta}_{\boldsymbol{\Theta}}; \boldsymbol{\phi}, \boldsymbol{\gamma})$ using the reparameterization trick.

---

**Algorithm 1** Semi-crowdsoursed clustering with DGMs (BayesSCDC)

---

**Input:** observations $\mathbf{O} = \{\mathbf{o}_1, ..., \mathbf{o}_N\}$, annotations $\mathbf{L}^{(1:M)}$, variational parameters $(\boldsymbol{\eta}_\Theta, \boldsymbol{\gamma}, \boldsymbol{\phi})$
**repeat**
    $\psi_i \leftarrow \langle r(\mathbf{o}_i; \boldsymbol{\phi}), \mathbf{t}(\mathbf{x}_i) \rangle, i = 1, ..., N$
    **for** each local variational parameter $\boldsymbol{\eta}^*_{\mathbf{x}_i}$ and $\boldsymbol{\eta}^*_{\mathbf{z}_i}$ **do**
       Update alternatively using eq. (12) and eq. (14)
    **end for**
    Sample $\hat{\mathbf{x}}_i \sim q^*(\mathbf{x}_i), i = 1, ..., N$
    Use $\hat{\mathbf{x}}_i$ to approximate $\mathbb{E}_{q^*(\mathbf{x})} \log p(\mathbf{o}|\mathbf{x}; \boldsymbol{\gamma})$ in the lower bound $\mathcal{J}$ eq. (16)
    Update the global variational parameters $\boldsymbol{\eta}_\Theta$ using the natural gradient in eq. (17)
    Update $\boldsymbol{\phi}, \boldsymbol{\gamma}$ using $\nabla_{\boldsymbol{\phi}, \boldsymbol{\gamma}} \mathcal{J}(\boldsymbol{\eta}_\Theta; \boldsymbol{\phi}, \boldsymbol{\gamma})$
**until** Convergence

---

**Stochastic approximation:** Computing the full natural gradient in eq. (17) requires to scan over all data and annotations, which is time-consuming. Similar to Section 2.3, we can approximate the variational lower bound with unbiased estimates using mini-batches of data and annotations, thus getting a stochastic natural gradient. Several sampling strategies have been developed for relational model [9] to keep the stochastic gradient unbiased. Here we choose the simplest way: we sample annotated data pairs uniformly from the annotations and form a subsample of the relational model, and do local message passing (eqs. (12) and (14)), then perform the global update using stochastic natural gradient calculated in the subsample. Besides, for all the unannotated data, we also subsample mini-batches from them and perform local and global steps without relational terms. The algorithm of BayesSCDC is shown in Algorithm 1.

**Comparison with SCDC** BayesSCDC is different in two aspects: (a) fully Bayesian treatment of global parameters; (b) variational algorithms. As we shall see in experiments, the result of (a) is that BayesSCDC can automatically determine the number of mixture components during training. As for (b), note that the variational family used in SCDC is not more flexible, but more restricted compared to BayesSCDC. In BayesSCDC, the mean-field $q(\mathbf{z})q(\mathbf{x})$ doesn't imply that $q^*(\mathbf{z})$ and $q^*(\mathbf{x})$ are independent, instead they implicitly influence each other through message passing in Eqs. (12) and (14). More importantly, in BayesSCDC the variational posterior gathers information from $\mathbf{L}$ through message passing in the relational model. In contrast, the amortized form $q(\mathbf{z}|\mathbf{o})q(\mathbf{x}, \mathbf{z}|\mathbf{o})$ used in SCDC ignores the effect of observed annotations $\mathbf{L}$. Another advantage of the inference algorithm in BayesSCDC is in the computational cost. As we have seen in Algorithm 1, the number of passes through the $\mathbf{x}$ to $\mathbf{o}$ network is no longer linear with $K$ because we get rid of summing over $\mathbf{z}$ in the observation term as in Section 2.3.

## 4 Related work

Most previous works on learning-from-crowds are about aggregating noisy crowdsourced labels from several predefined classes [5, 18, 26, 32, 24]. A common way they use is to simultaneously estimate the workers' behavior models and the ground truths. Different from this line of work, crowdclustering [8] collects pairwise labels, including the must-links and the cannot-links, from the crowds, then discovers the items' affiliations as well as the category structure from these noisy labels, so it can be used on a border range of applications compared with the classification methods. Recent work [25] also developed crowdclustering algorithm on triplet annotations.

One shortcoming of crowdclustering is that it can only cluster objects with available manual annotations. For large-scale problems, it is not feasible to have each object manually annotated by multiple workers. Similar problems were extensively discussed in the semi-supervised clustering area, where we are given the features for all the items and constraints on only a same portion of the items. Metric learning methods, including Information-Theoretic Metric Learning (**ITML**) [4] and Metric Pairwise Constrained KMeans (**MPCKMeans**) [1], are used on this problem, they first learn the similarity metric between items mainly based on the supervised portion of data, then cluster the rest items using this metric. Semi-crowdsourced clustering (**SemiCrowd**) [31] combines the idea of crowdclustering and semi-supervised clustering, it aims to learn a pairwise similarity measure from the crowdsourced labels of $n$ objects ($n \ll N$) and the features of $N$ objects. Unlike crowdclustering, the number of clusters in SemiCrowd is assumed to be given a priori. And it doesn't estimate the behavior

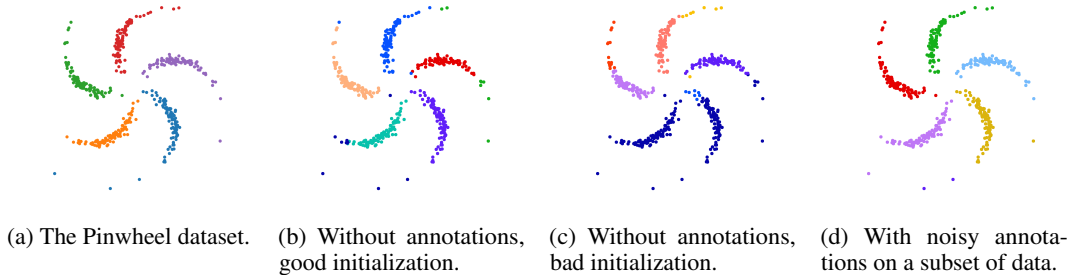

(a) The Pinwheel dataset.   (b) Without annotations, good initialization.   (c) Without annotations, bad initialization.   (d) With noisy annotations on a subset of data.

Figure 2: Clustering results on the Pinwheel dataset, with each color representing one cluster.

of different workers. Multiple Clustering Views from the Crowd (**MCVC**) [3] extends the idea to discover several different clustering results from the noisy labels provided by uncertain experts. A common shortcoming of these semi-crowdsourced clustering methods is they cannot make good use of unlabeled items when measuring the similarities, while our model is a step towards this direction.

As shown in Section 2.1, our model is a deep generative model (DGM) with relational latent structures. DGMs are a kind of probabilistic graphical models that use neural networks to parameterize the conditional distribution between random variables. Unlike traditional probabilistic models, DGMs can directly model high-dimensional outputs with complex structures, which enables end-to-end training on real data. They have shown success in (conditional) image generation [15, 19], semi-supervised learning [14], and one-shot classification [20]. Typical inference algorithms for DGMs are in the amortized form like that in Section 2.3. However, this approach cannot leverage the conjugate structures in latent variables. Therefore few works have been done on fully Bayesian treatment of global parameters in DGMs. [13, 16] are two exceptions. In [13] the authors propose using recognition networks to produce conjugate graphical model potentials, so that traditional variational message passing algorithms and natural gradient updates can be easily combined with amortized learning of network parameters. Our work extends their algorithm to relational observations, which has not been investigated before.

## 5  Experiments

In this section, we demonstrate the effectiveness of the proposed methods on synthetic and real-world datasets with simulated or crowdsourced noisy annotations. Code is available at `https://github.com/xinmei9322/semicrowd`. Part of the implementation is based on ZhuSuan [22].

### 5.1  Toy Pinwheel dataset

**Simulating noisy annotations from workers.** Suppose we have $M$ workers with accuracy parameters $(\alpha_m, \beta_m)$. We random sample pairs of items $\mathbf{o}_i$ and $\mathbf{o}_j$ and generate the annotations provided by worker $m$ based on the true clustering labels of $\mathbf{o}_i$ and $\mathbf{o}_j$ as well as the worker's accuracy parameters $(\alpha_m, \beta_m)$. If $\mathbf{o}_i$ and $\mathbf{o}_j$ belong to the same cluster, the worker has probability $\alpha_m$ to provide ML constraint $L_{ij}^{(m)} = 1$. If not, the worker has probability $\beta_m$ to provide CL constraint $L_{ij}^{(m)} = 0$.

**Evaluation metrics.** The clustering performance is evaluated by the commonly used normalized mutual information (NMI) score [23], measuring the similarity between two partitions. Following recent work [29], we also report the unsupervised clustering accuracy, which requires to compute the best mapping using the Hungarian algorithm efficiently.

First we apply our method to a toy example–the pinwheel dataset in Fig. 2a following [13, 16]. It has 5 clusters and each cluster has 100 data points, thus there are 500 data points in total. We compare with unsupervised clustering to understand the help of noisy annotations. The clustering results are shown in Fig. 2. We random sampled 100 data points for annotations and simulate 20 workers, each worker gives 49 pairs of annotations, 980 in total. We set equal accuracy to each worker $\alpha_m = \beta_m = 0.9$.

We use the fully Bayesian model (BayesSCDC) described in Section 3. The initial number of clusters is set to a larger number $K = 15$ since the hyper priors have sparsity property intrinsically and can learn the number of clusters automatically. Unsupervised clustering is sensitive to the initializations, which achieves 95.6% accuracy and NMI score 0.91 with good initializations as shown in Fig. 2b.

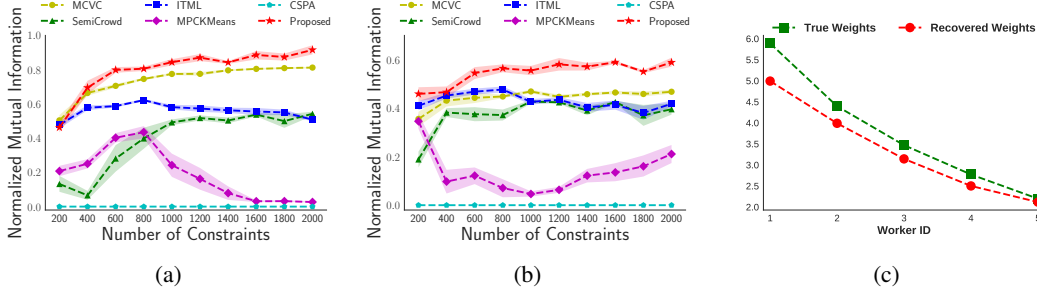

Figure 3: Comparison to baselines: (a) Face: All the data points are annotated; (b) Face: Only 100 data points are annotated; (c) True accuracies are set to $\boldsymbol{\alpha} = \boldsymbol{\beta} = [0.95, 0.9, 0.85, 0.8, 0.75]$. The green line is the true weights of each worker and the red line is the estimated weights by our model.

After training, it learns $K = 8$ clusters. However, with bad initializations, the accuracy and NMI score of unsupervised clustering are 75.6% and 0.806, respectively, as shown in Fig. 2c. With noisy annotations on random sampled 100 data points, our model improves accuracy to 96.6% and NMI score to 0.94. And it converges to $K = 6$ clusters. Our model prevents the bad results in Fig. 2c by making use of annotations.

## 5.2 UCI benchmark experiments

In this subsection, we compare the proposed SCDC with the competing methods on the UCI benchmarks. The baselines include **MCVC** [3], **SemiCrowd** [31], semi-supervised clustering methods such as **ITML** [4], **MPCKMeans** [1] and Cluster-based Similarity Partitioning Algorithm (**CSPA**) [23].

Crowdsourced annotations are not available for UCI datasets. Following the experimental protocol in **MCVC** [3], we generate noisy annotations given by $M = 5$ simulated workers with different sensitivity and specificity, i.e., $\boldsymbol{\alpha} = \boldsymbol{\beta} = [0.95, 0.9, 0.85, 0.8, 0.75]$, which is more challenging than equal accuracy parameters. The annotations provided by each worker varies from 200 to 2000 and the number of ML constraints equals to the number of CL constraints.

We test on Face dataset [7], containing 640 face images from 20 people with different poses (straight, left, right, up). The ground-truth clustering is based on the poses. The original image has 960 pixels. To speed up training, baseline methods apply Principle Component Analysis (PCA) and keep 20 components. For fair comparison, we test the proposed SCDC on the features after PCA. Fig. 3 plots the mean and standard deviation of NMI scores in 10 different runs for each fixed number of constraints. In Fig. 3a, the annotations are randomly generated on the whole dataset. We observe that our method consistently outperforms all competing methods, demonstrating that the clustering benefits from the joint generative modeling of inputs and annotations.

**Annotations on a subset.** To illustrate the benefits of our method in the situation where only a small part of data points are annotated, we simulate noisy annotations on only 100 images. Fig. 3b shows the results of 100 annotated images. Our method exploits more structure information in the unlabeled data and shows notable improvements over all competing methods.

**Recover worker behaviors.** For each worker $m$, our model estimates the different accuracies $\alpha_m$ and $\beta_m$. We can derive from eq. (2) that the annotations of each worker $m$ are weighted by $\log \frac{\alpha_m}{1-\alpha_m} + \log \frac{\beta_m}{1-\beta_m}$, which means workers with higher accuracies are more reliable and will be weighted higher. We plot the weights of 5 workers in the Face experiments in Fig. 3c.

## 5.3 End-to-end training with raw images

**MNIST** As mentioned earlier, an important feature of DGMs is that they can directly model raw data, such as images. To verify this, we experiment with the MNIST dataset of digit images, which includes 60k training images from handwritten digits 0-9. We collect crowdsourced annotations from $M = 48$ workers and get 3276 annotations in total. The two variants of our model (SCDC, BayesSCDC) are tested with or without annotations. For BayesSCDC, a non-informative prior $\text{Beta}(1, 1)$ is placed over $\boldsymbol{\alpha}, \boldsymbol{\beta}$. For fair comparison, we also randomly sample the initial accuracy parameters $\boldsymbol{\alpha}, \boldsymbol{\beta}$ from $\text{Beta}(1, 1)$ for SCDC. We average the results of 5 runs. In each run we randomly initialize the model for 10 times and pick the best result. All models are trained for

Table 1: Clustering performance on MNIST. The average time per epoch is reported.

| Method | without annotations | | | with annotations | | |
|---|---|---|---|---|---|---|
| | **Accuracy** | **NMI** | **Time** | **Accuracy** | **NMI** | **Time** |
| SCDC | $65.92 \pm 3.47$ % | $0.6953 \pm 0.0167$ | 177.3s | $81.87 \pm 3.86$% | $0.7657 \pm 0.0233$ | 201.7s |
| BayesSCDC | $77.64 \pm 3.97$ % | $0.7944 \pm 0.0178$ | 11.2s | $\mathbf{84.24 \pm 5.52\%}$ | $\mathbf{0.8120 \pm 0.0210}$ | 16.4s |

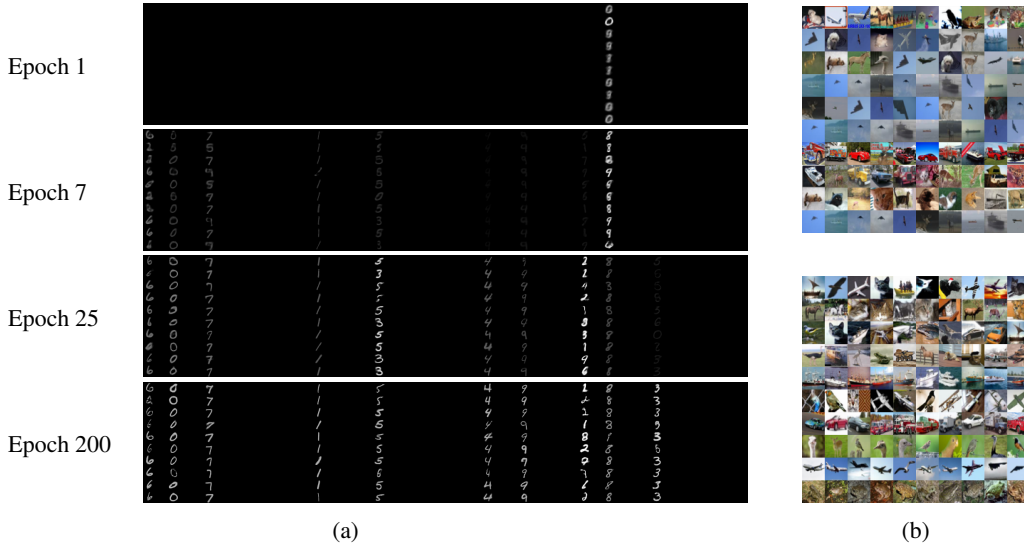

(a)                                                                (b)

Figure 4: (a) MNIST: visualization of generated random samples of 50 clusters during training BayesSCDC. Each column represents a cluster, whose inferred proportion ($\pi_k$) is reflected by brightness; (b) Clustering results on CIFAR-10: (top) unsupervised; (bottom) with noisy annotations.

200 epochs with minibatch size of 128 for each random initialization. The results are shown in Table 1. We can see that both models can effectively combine the information from the raw data and annotations, i.e., they worked reasonably well with only unlabeled data, and better when given noisy annotations on a subset of data. In terms of clustering accuracy and NMI, BayesSCDC outperforms SCDC. We believe that this is because the variational message passing algorithm used in BayesSCDC can effectively gather information from the crowdsourced annotations to form better variational approximations, as explained in Section 3.2. Besides being more accurate, BayesSCDC is much faster because the computation cost caused by neural networks does not scales linearly with the number of clusters $K$ (50 in this case). In Fig. 4a we show that BayesSCDC is more flexible and automatically determines the number of mixture components during training.

**CIFAR-10**   We also conduct experiments with real crowdsourced labels on more complex natural images, i.e., CIFAR-10. Using the same crowdsourcing scheme, we collect 8640 noisy annotations from 32 web workers on a subset of randomly sampled 4000 images. We apply SCDC with/without annotations for 5 runs of random initializations. SCDC without annotations failed with NMI score $0.0424 \pm 0.0119$ and accuracy $14.23 \pm 0.69$% among 5 runs. But the NMI score achieved by SCDC with noisy annotations is $0.5549 \pm 0.0028$ and the accuracy is $50.09 \pm 0.08$%. The clustering results on test dataset are shown in Fig. 4b. We plot 10 test samples with the largest probability for each cluster. More experiment details and discussions could be found in the supplementary material.

## 6   Conclusion

In this paper, we proposed a semi-crowdsourced clustering model based on deep generative models and its fully Bayesian version. We developed fast (natural-gradient) stochastic variational inference algorithms for them. The resulting method can jointly model the crowdsourced labels, worker behaviors, and the (un)annotated items. Experiments have demonstrated that the proposed method outperforms previous competing methods on standard benchmark datasets. Our work also provides general guidelines on how to incorporate DGMs to statistical relational models, where the proposed inference algorithm can be applied under a broader context.

## Acknowledgement

Yucen Luo would like to thank Matthew Johnson for helpful discussions on the SVAE algorithm [13], and Yale Chang for sharing the code of the UCI benchmark experiments. We thank the anonymous reviewers for feedbacks that greatly improved the paper. This work was supported by the National Key Research and Development Program of China (No. 2017YFA0700904), NSFC Projects (Nos. 61620106010, 61621136008, 61332007), Beijing NSF Project (No. L172037), Tiangong Institute for Intelligent Computing, NVIDIA NVAIL Program, and the projects from Siemens, NEC and Intel.

## Footnotes

[2]Detailed expressions of each distribution can be found in Appendix A.

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
