[Supplementary Material]

# A Derivations

## A.1 Natural parameters and sufficient statistics

$$\boldsymbol{\eta_\pi} = \begin{bmatrix} \alpha_1 - 1 \\ \vdots \\ \alpha_K - 1 \end{bmatrix}, \boldsymbol{\eta_z}(\boldsymbol{\pi}) = \begin{bmatrix} \log \pi_1 \\ \vdots \\ \log \pi_K \end{bmatrix}, (\boldsymbol{\eta_{\mu,\Sigma}})^{(k)} = \begin{bmatrix} \kappa \mathbf{m} \\ \mathrm{vec}(\mathbf{S} + \kappa \mathbf{mm}^\top) \\ \kappa \\ \nu + d + 2 \end{bmatrix}, \boldsymbol{\eta_x}(\boldsymbol{\mu}, \boldsymbol{\Sigma}) = \begin{bmatrix} \boldsymbol{\Sigma}^{-1}\boldsymbol{\mu} \\ \mathrm{vec}(-\frac{1}{2}\boldsymbol{\Sigma}^{-1}) \end{bmatrix}.$$

$$\mathbf{t}(\boldsymbol{\pi}) = \begin{bmatrix} \log \pi_1 \\ \vdots \\ \log \pi_K \end{bmatrix}, \mathbf{t}(\mathbf{z}) = \begin{bmatrix} z_1 \\ \vdots \\ z_K \end{bmatrix}, \mathbf{t}(\boldsymbol{\mu}, \boldsymbol{\Sigma})^{(k)} = \begin{bmatrix} \boldsymbol{\Sigma}_k^{-1}\boldsymbol{\mu}_k \\ \mathrm{vec}(-\frac{1}{2}\boldsymbol{\Sigma}_k^{-1}) \\ -\frac{1}{2}\boldsymbol{\mu}_k^\top \boldsymbol{\Sigma}_k^{-1}\boldsymbol{\mu}_k \\ -\frac{1}{2}\ln|\boldsymbol{\Sigma}_k| \end{bmatrix}, \mathbf{t}(\mathbf{x}) = \begin{bmatrix} \mathbf{x} \\ \mathrm{vec}(\mathbf{xx}^\top) \end{bmatrix}.$$

$$(\boldsymbol{\eta_\alpha})^{(m)} = \begin{bmatrix} \tau_{\alpha_m^1} - 1 \\ \tau_{\alpha_m^2} - 1 \end{bmatrix}, \quad \mathbf{t}(\boldsymbol{\alpha})^{(m)} = \begin{bmatrix} \log \alpha_m \\ \log(1 - \alpha_m) \end{bmatrix}, \quad \boldsymbol{\eta_\beta} \text{ and } \mathbf{t}(\boldsymbol{\beta}) \text{ are similar.}$$

Note that $\mathbf{t}(\boldsymbol{\pi}) = \boldsymbol{\eta_z}(\boldsymbol{\pi})$, $\boldsymbol{\eta_x}(\boldsymbol{\mu}, \boldsymbol{\Sigma}) = \mathbf{t}(\boldsymbol{\mu}, \boldsymbol{\Sigma})[: 2]$, where the conjugacy comes. In practice, we parameterized the unnormalized version of $\tilde{\boldsymbol{\eta}}_z(\boldsymbol{\pi}) = \begin{bmatrix} \log \pi_1 + c \\ \vdots \\ \log \pi_K + c \end{bmatrix}$, since it is unconstrained. The update rule for $\tilde{\boldsymbol{\eta}}_z(\boldsymbol{\pi})$ is the same with $\boldsymbol{\eta_z}(\boldsymbol{\pi})$ due to their constant difference.

## A.2 Expected sufficient statistics

$$\mathbb{E}_{q(\boldsymbol{\pi})}\mathbf{t}(\boldsymbol{\pi}) = \begin{bmatrix} \psi(\alpha_1) \\ \vdots \\ \psi(\alpha_K) \end{bmatrix} - \psi\left(\sum_{k=1}^{K} \alpha_k\right), \quad \mathbb{E}_{q(\boldsymbol{\mu},\boldsymbol{\Sigma})}\mathbf{t}(\boldsymbol{\mu}, \boldsymbol{\Sigma})^{(k)} = \begin{bmatrix} \nu \mathbf{S}^{-1}\mathbf{m} \\ \mathrm{vec}(-\frac{1}{2}\nu \mathbf{S}^{-1}) \\ -\frac{1}{2}(\kappa^{-1}d + \nu \mathbf{m}^\top \mathbf{S}^{-1}\mathbf{m}) \\ \frac{1}{2}\left(\psi_d(\frac{\nu}{2}) + d\ln 2 - \ln|\mathbf{S}|\right) \end{bmatrix},$$

where $\psi_d(\frac{\nu}{2}) = \sum_{i=1}^{d} \psi(\frac{\nu+1-i}{2})$, $\psi$ is the digamma function.

$$\mathbb{E}_{q(\mathbf{z})}\mathbf{t}(\mathbf{z}) = \begin{bmatrix} \pi_1 \\ \vdots \\ \pi_K \end{bmatrix}, \quad \mathbb{E}_{q(\mathbf{x})}\mathbf{t}(\mathbf{x}) = \begin{bmatrix} \boldsymbol{\mu}, \\ \mathrm{vec}(\boldsymbol{\Sigma} + \boldsymbol{\mu}\boldsymbol{\mu}^\top) \end{bmatrix}$$

$$\mathbb{E}_{q(\boldsymbol{\alpha})}\mathbf{t}(\boldsymbol{\alpha})^{(m)} = \begin{bmatrix} \psi(\tau_{\alpha_m^1}) \\ \psi(\tau_{\alpha_m^2}) \end{bmatrix} - \psi(\tau_{\alpha_m^1} + \tau_{\alpha_m^2}), \quad \mathbb{E}_{q(\boldsymbol{\beta})}\mathbf{t}(\boldsymbol{\beta}) \text{ is similar.}$$

## A.3 Log partition function

$$\ln Z(\boldsymbol{\eta_\pi}) = \sum_{k=1}^{K} \ln \Gamma(\alpha_k) - \ln \Gamma\left(\sum_{k=1}^{K} \alpha_k\right), \quad \ln Z(\boldsymbol{\eta_{\mu,\Sigma}}) = \frac{\nu}{2}(d\ln 2 - \ln|S|) + \ln \Gamma_d\left(\frac{\nu}{2}\right) - \frac{d}{2}\ln \kappa,$$

where $\Gamma_d(\frac{\nu}{2}) = \pi^{d(d-1)/4} \prod_{i=1}^{d} \Gamma\left(\frac{\nu+1-i}{2}\right)$, $\Gamma$ is the Gamma function.

$$\ln Z(\boldsymbol{\eta_z}(\boldsymbol{\pi})) = 0, \quad \ln Z(\boldsymbol{\eta_x}(\boldsymbol{\mu}, \boldsymbol{\Sigma})) = \frac{1}{2}\left(\boldsymbol{\mu}^\top \boldsymbol{\Sigma}^{-1}\boldsymbol{\mu} + \ln|\boldsymbol{\Sigma}|\right) = -\mathbf{1}^\top \{\mathbf{t}(\boldsymbol{\mu}, \boldsymbol{\Sigma})[2 :]\}.$$

$$\ln Z(\boldsymbol{\eta_\alpha}) = \ln \Gamma(\tau_{\alpha^1}) + \ln \Gamma(\tau_{\alpha^2}) - \ln \Gamma(\tau_{\alpha^1} + \tau_{\alpha^2}).$$

## A.4 Variational message passing for local parameters

The local update for $\mathbf{x}_i$:

$$\boldsymbol{\eta}_{\mathbf{x}_i}^* = \mathbb{E}_{q(\boldsymbol{\mu},\boldsymbol{\Sigma})}[\mathbf{t}(\boldsymbol{\mu},\boldsymbol{\Sigma})[:2]]^\top \mathbb{E}_{q(\mathbf{z}_i)}\mathbf{t}(\mathbf{z}_i) + r(\mathbf{o}_i;\boldsymbol{\phi}). \tag{18}$$

The local update for $\mathbf{z}_i$:

$$\boldsymbol{\eta}_{\mathbf{z}_i}^* = \mathbb{E}_{q(\boldsymbol{\pi})}\mathbf{t}(\boldsymbol{\pi}) + \mathbb{E}_{q(\boldsymbol{\mu},\boldsymbol{\Sigma})}\left[\mathbf{t}(\boldsymbol{\mu},\boldsymbol{\Sigma})\right]^\top \mathbb{E}_{q(\mathbf{x}_i)}\left[(\mathbf{t}(\mathbf{x}_i),\mathbf{1})\right] + \sum_{m=1}^{M}\sum_{j=1}^{N} w_{ij}^{(m)}\mathbb{E}_{q(\mathbf{z}_j)}[\mathbf{t}(\mathbf{z}_j)].$$

where $w_{ij}^{(m)} = I_{ij}^{(m)}\mathbb{E}_{q(\boldsymbol{\alpha},\boldsymbol{\beta})}\left[\ln\frac{1-\alpha_m}{\beta_m} + L_{ij}^{(m)}\left(\ln\frac{\alpha_m}{1-\alpha_m} + \ln\frac{\beta_m}{1-\beta_m}\right)\right].$

## A.5 The final objective

The final objective:

$$\begin{aligned}
\mathcal{J}(\boldsymbol{\eta}_{\boldsymbol{\Theta}};\boldsymbol{\phi},\boldsymbol{\gamma}) &\triangleq \mathcal{L}(\boldsymbol{\eta}_{\boldsymbol{\Theta}},\boldsymbol{\eta}_{\mathbf{Z}}^*(\boldsymbol{\eta}_{\boldsymbol{\Theta}},\boldsymbol{\phi}),\boldsymbol{\eta}_{\mathbf{X}}^*(\boldsymbol{\eta}_{\boldsymbol{\Theta}},\boldsymbol{\phi}),\boldsymbol{\gamma}) \\
&= \mathbb{E}_{q^*(\mathbf{X})}\log p(\mathbf{O}|\mathbf{X};\boldsymbol{\gamma}) + \mathbb{E}_{q(\boldsymbol{\alpha},\boldsymbol{\beta})q^*(\mathbf{Z})}\log p(\mathbf{L}^{(1:M)}|\mathbf{Z},\boldsymbol{\alpha},\boldsymbol{\beta}) \\
&\quad - \mathbb{E}_{q(\boldsymbol{\pi},\boldsymbol{\mu},\boldsymbol{\Sigma})}\mathrm{KL}(q^*(\mathbf{Z})q^*(\mathbf{X})\|p(\mathbf{Z}|\boldsymbol{\pi})p(\mathbf{X}|\boldsymbol{\mu},\boldsymbol{\Sigma},\mathbf{Z})) \\
&\quad - \mathrm{KL}(q(\boldsymbol{\pi},\boldsymbol{\mu},\boldsymbol{\Sigma},\boldsymbol{\alpha},\boldsymbol{\beta})\|p(\boldsymbol{\pi},\boldsymbol{\mu},\boldsymbol{\Sigma},\boldsymbol{\alpha},\boldsymbol{\beta})).
\end{aligned} \tag{19}$$

The annotation likelihood term:

$$\mathbb{E}_{q(\boldsymbol{\alpha},\boldsymbol{\beta})q^*(\mathbf{Z})}\log p(\mathbf{L}^{(1:M)}|\mathbf{Z},\boldsymbol{\alpha},\boldsymbol{\beta}) =$$
$$\sum_{m=1}^{M}\sum_{1\le i<j\le N}\left[w_{ij}^{(m)}\mathbb{E}_{q(\mathbf{z}_i)}[\mathbf{t}(\mathbf{z}_i)]^\top\mathbb{E}_{q(\mathbf{z}_j)}\mathbf{t}(\mathbf{z}_j) + I_{ij}^{(m)}\mathbb{E}_{q(\boldsymbol{\beta})}\left(L_{ij}^{(m)}\ln\frac{1-\beta_m}{\beta_m} + \ln\beta_m\right)\right]. \tag{20}$$

The local KL divergence term:

$$\begin{aligned}
&\mathbb{E}_{q(\boldsymbol{\pi},\boldsymbol{\mu},\boldsymbol{\Sigma})}\mathrm{KL}(q^*(\mathbf{Z})q^*(\mathbf{X})\|p(\mathbf{Z}|\boldsymbol{\pi})p(\mathbf{X}|\boldsymbol{\mu},\boldsymbol{\Sigma},\mathbf{Z})) \\
&= \mathbb{E}_{q(\boldsymbol{\pi})}\mathrm{KL}(q^*(\mathbf{Z})\|p(\mathbf{Z}|\boldsymbol{\pi})) + \mathbb{E}_{q(\boldsymbol{\mu},\boldsymbol{\Sigma})q^*(\mathbf{Z})}\mathrm{KL}(q^*(\mathbf{X})\|p(\mathbf{X}|\boldsymbol{\mu},\boldsymbol{\Sigma},\mathbf{Z})) \\
&= \sum_{i=1}^{N}\left\{\mathbb{E}_{q(\boldsymbol{\pi})}\mathrm{KL}(q^*(\mathbf{z}_i)\|p(\mathbf{z}_i|\boldsymbol{\pi})) + \mathbb{E}_{q(\boldsymbol{\mu},\boldsymbol{\Sigma})q^*(\mathbf{z}_i)}\mathrm{KL}(q^*(\mathbf{x}_i)\|p(\mathbf{x}_i|\boldsymbol{\mu},\boldsymbol{\Sigma},\mathbf{z}_i))\right\}. \tag{21}
\end{aligned}$$

$$\begin{aligned}
\mathrm{KL}(q^*(\mathbf{z}_i)\|p(\mathbf{z}_i|\boldsymbol{\pi})) &= \langle\boldsymbol{\eta}_{\mathbf{z}_i}^* - \boldsymbol{\eta}_{\mathbf{z}_i}(\boldsymbol{\pi}), \mathbb{E}_{q^*(\mathbf{z}_i)}\mathbf{t}(\mathbf{z}_i)\rangle \\
\mathbb{E}_{q(\boldsymbol{\pi})}\mathrm{KL}(q^*(\mathbf{z}_i)\|p(\mathbf{z}_i|\boldsymbol{\pi})) &= \langle\boldsymbol{\eta}_{\mathbf{z}_i}^* - \mathbb{E}_{q(\boldsymbol{\pi})}\mathbf{t}(\boldsymbol{\pi}), \mathbb{E}_{q^*(\mathbf{z}_i)}\mathbf{t}(\mathbf{z}_i)\rangle.
\end{aligned}$$

$$\mathrm{KL}(q^*(\mathbf{x}_i)\|p(\mathbf{x}_i|\boldsymbol{\mu}_k,\boldsymbol{\Sigma}_k)) = \langle\boldsymbol{\eta}_{\mathbf{x}_i}^* - \boldsymbol{\eta}_{\mathbf{x}_i}(\boldsymbol{\mu},\boldsymbol{\Sigma})^{(k)}, \mathbb{E}_{q^*(\mathbf{x}_i)}\mathbf{t}(\mathbf{x}_i)\rangle - \left[\ln Z(\boldsymbol{\eta}_{\mathbf{x}_i}^*) - \ln Z(\boldsymbol{\eta}_{\mathbf{x}_i}(\boldsymbol{\mu},\boldsymbol{\Sigma})^{(k)})\right] \tag{22}$$

$$\begin{aligned}
\mathbb{E}_{q(\boldsymbol{\mu},\boldsymbol{\Sigma})q^*(\mathbf{z}_i)}\mathrm{KL}(q^*(\mathbf{x}_i)\|p(\mathbf{x}_i|\boldsymbol{\mu},\boldsymbol{\Sigma},\mathbf{z}_i)) &= \langle\boldsymbol{\eta}_{\mathbf{x}_i}^* - \mathbb{E}_{q(\boldsymbol{\mu},\boldsymbol{\Sigma})}[\boldsymbol{\eta}_{\mathbf{x}_i}(\boldsymbol{\mu},\boldsymbol{\Sigma})]^\top\mathbb{E}_{q^*(\mathbf{z}_i)}\mathbf{t}(\mathbf{z}_i), \mathbb{E}_{q^*(\mathbf{x}_i)}\mathbf{t}(\mathbf{x}_i)\rangle \\
&\quad - \left\{\ln Z(\boldsymbol{\eta}_{\mathbf{x}_i}^*) - \mathbb{E}_{q(\boldsymbol{\mu},\boldsymbol{\Sigma})}[\ln Z(\boldsymbol{\eta}_{\mathbf{x}_i}(\boldsymbol{\mu},\boldsymbol{\Sigma}))]^\top\mathbb{E}_{q*(\mathbf{z}_i)}\mathbf{t}(\mathbf{z}_i)\right\} \\
&= \langle\boldsymbol{\eta}_{\mathbf{x}_i}^* - \mathbb{E}_{q(\boldsymbol{\mu},\boldsymbol{\Sigma})}[\mathbf{t}(\boldsymbol{\mu},\boldsymbol{\Sigma})[:2]]^\top\mathbb{E}_{q^*(\mathbf{z}_i)}\mathbf{t}(\mathbf{z}_i), \mathbb{E}_{q^*(\mathbf{x}_i)}\mathbf{t}(\mathbf{x}_i)\rangle \\
&\quad - \left\{\ln Z(\boldsymbol{\eta}_{\mathbf{x}_i}^*) + \mathbf{1}^\top\mathbb{E}_{q(\boldsymbol{\mu},\boldsymbol{\Sigma})}[\mathbf{t}(\boldsymbol{\mu},\boldsymbol{\Sigma})[2:]]^\top\mathbb{E}_{q*(\mathbf{z}_i)}\mathbf{t}(\mathbf{z}_i)\right\}.
\end{aligned}$$

The global KL divergence term:

$$\mathrm{KL}(q(\boldsymbol{\pi})\|p(\boldsymbol{\pi})) = \langle\boldsymbol{\eta}_{\boldsymbol{\pi}} - \boldsymbol{\eta}_{\boldsymbol{\pi}}^0, \mathbb{E}_{q(\boldsymbol{\pi})}\mathbf{t}(\boldsymbol{\pi})\rangle - \left[\ln Z(\boldsymbol{\eta}_{\boldsymbol{\pi}}) - \ln Z(\boldsymbol{\eta}_{\boldsymbol{\pi}}^0)\right]. \tag{23}$$

$$\begin{aligned}
\mathrm{KL}(q(\boldsymbol{\mu},\boldsymbol{\Sigma})\|p(\boldsymbol{\mu},\boldsymbol{\Sigma})) = \sum_{k=1}^{K}\Big\{&\langle\boldsymbol{\eta}_{\boldsymbol{\mu},\boldsymbol{\Sigma}}^{(k)} - \boldsymbol{\eta}_{\boldsymbol{\mu},\boldsymbol{\Sigma}}^{0\,(k)}, \mathbb{E}_{q(\boldsymbol{\mu},\boldsymbol{\Sigma})}\mathbf{t}(\boldsymbol{\mu},\boldsymbol{\Sigma})^{(k)}\rangle \\
&- \left[\ln Z(\boldsymbol{\eta}_{\boldsymbol{\mu},\boldsymbol{\Sigma}}^{(k)}) - \ln Z(\boldsymbol{\eta}_{\boldsymbol{\mu},\boldsymbol{\Sigma}}^{0\,(k)})\right]\Big\}. \tag{24}
\end{aligned}$$

$$\mathrm{KL}(q(\boldsymbol{\alpha})\|p(\boldsymbol{\alpha})) = \langle \boldsymbol{\eta_\alpha} - \boldsymbol{\eta_\alpha^0}, \mathbb{E}_{q(\boldsymbol{\alpha})}\mathbf{t}(\boldsymbol{\alpha})\rangle - \left[\ln Z(\boldsymbol{\eta_\alpha}) - \ln Z(\boldsymbol{\eta_\alpha^0})\right], \quad \boldsymbol{\beta} \text{ is similar.} \tag{25}$$

Global variational parameter updates:

$$\tilde{\nabla}_{\boldsymbol{\eta_\pi}}\mathcal{J} \approx \boldsymbol{\eta_\pi^0} + \sum_{i=1}^N \mathbb{E}_{q(\mathbf{z}_i)}\mathbf{t}(\mathbf{z}_i) - \boldsymbol{\eta_\pi}, \tag{26}$$

$$\tilde{\nabla}_{\boldsymbol{\eta_{\mu,\Sigma}}}\mathcal{J} \approx \boldsymbol{\eta_{\mu,\Sigma}^0} + \sum_{i=1}^N \left[\mathbb{E}_{q(\mathbf{x}_i)}(\mathbf{t}(\mathbf{x}_i),\mathbf{1})\right]^\top \mathbb{E}_{q(\mathbf{z}_i)}\mathbf{t}(\mathbf{z}_i) - \boldsymbol{\eta_{\mu,\Sigma}}, \tag{27}$$

$$\tilde{\nabla}_{\boldsymbol{\eta_\alpha^{(m)}}}\mathcal{J} = (\boldsymbol{\eta_\alpha^0})^{(m)} + \frac{1}{2}\sum_{i,j=1}^N I_{ij}^{(m)}\mathbb{E}_{q(\mathbf{z}_i)}[\mathbf{t}(\mathbf{z}_i)]^\top \mathbb{E}_{q(\mathbf{z}_j)}\mathbf{t}(\mathbf{z}_j)\begin{bmatrix} L_{ij}^{(m)} \\ 1 - L_{ij}^{(m)} \end{bmatrix} - \boldsymbol{\eta_\alpha^{(m)}}, \tag{28}$$

$$\tilde{\nabla}_{\boldsymbol{\eta_\beta^{(m)}}}\mathcal{J} = (\boldsymbol{\eta_\beta^0})^{(m)} + \frac{1}{2}\sum_{i,j=1}^N I_{ij}^{(m)}\left(1 - \mathbb{E}_{q(\mathbf{z}_i)}[\mathbf{t}(\mathbf{z}_i)]^\top \mathbb{E}_{q(\mathbf{z}_j)}\mathbf{t}(\mathbf{z}_j)\right)\begin{bmatrix} 1 - L_{ij}^{(m)} \\ L_{ij}^{(m)} \end{bmatrix} - \boldsymbol{\eta_\beta^{(m)}}. \tag{29}$$

## B  Experiment settings

**Toy Pinwheel dataset**    We follow the synthetic data generation process and hyperparameter settings in [13]. The latent dim of $\mathbf{x}$ is set to $d = 2$ and initial number of clusters $K = 15$. The Dirichlet prior for mixing coefficients $\boldsymbol{\pi}$: $\alpha_0 = 0.05/K$. The NIW prior $p(\boldsymbol{\mu}, \boldsymbol{\Sigma})$: concentration $\kappa = 0.5$, location parameter $\boldsymbol{m} = \mathbf{0}$, scale matrix $\mathbf{S} = (d + \kappa)\mathbf{I}_d$ ($\mathbf{I}_d \in \mathbb{R}^{d \times d}$ is the identity matrix), degrees of freedom $\nu = d + \kappa$. The Beta prior hyperparameter $\tau_{\alpha_0^1}, \tau_{\alpha_0^2}, \tau_{\beta_0^1}, \tau_{\beta_0^2}$ for accuracy $(\boldsymbol{\alpha}, \boldsymbol{\beta})$ are all set to 1. The hyperparameters in variational distributions of global variables are $\alpha_0 \sim \mathcal{U}(1,2), \kappa = 1, \boldsymbol{m} \sim \mathcal{N}(0,3), \mathbf{S} = (d + \kappa)\mathbf{I}_d, \nu = d + \kappa, \tau_{\alpha_0^1} = \tau_{\beta_0^1} = 10, \tau_{\alpha_0^2} = \tau_{\beta_0^2} = 1$. The networks of $p(\mathbf{o}|\mathbf{x}; \boldsymbol{\gamma})$ and recognition networks $r(\mathbf{o}; \boldsymbol{\phi})$ are both MLPs with two hidden layers of 40 ReLU units. The networks are trained for 20 epochs with minibatch size $|B|$ of 50, momentum 0.9 and learning rate $1 \times 10^{-3}$. The learning rate for global variational parameters $\boldsymbol{\eta_\Theta}$ is $1 \times 10^{-4}$. The batch size $|S|$ of annotations is computed by $|S| = N_a \times |B|/N$ to ensure that the annotations and the images are feed to the network proportionally.

**UCI dataset**    The implementation of competing methods are based on the source code provided by MCVC [3]. In our proposed SCDC, the preprocessing of images and generation of annotations are the same as that in MCVC. The latent dim is set to $d = 8$ and $K = 4$. As for the network architecture, $p(\mathbf{o}|\mathbf{x}; \boldsymbol{\gamma})$ and recognition networks $q(\mathbf{z}_n|\mathbf{o}_n; \boldsymbol{\phi})q(\mathbf{x}_n|\mathbf{z}_n, \mathbf{o}_n; \boldsymbol{\phi})$ are simple MLPs with two hidden layers of 40 ReLU units. The networks are trained for 20 epochs using Adam optimizer with learning rate $1 \times 10^{-3}$ and batch size $|B| = 10$ .

**MNIST**    For both SCDC and BayesSCDC, $p(\mathbf{o}|\mathbf{x})$ and recognition networks are MLPs with two hidden layers of 500 ReLU units. The latent dim of $\mathbf{x}$ is $d = 8$ and initial number of clusters $K = 50$. Both two models are trained for 200 epochs. For SCDC, we use Adam optimizer with learning rate $1 \times 10^{-3}$ and minibatch size 128. For BayesSCDC, the hyperparameter settings in the priors and variational distributions are all the same as that in the Pinwheel dataset. The networks are trained with minibatch size $|B|$ of 128, momentum 0.9 and learning rate $1 \times 10^{-3}$.

**CIFAR-10**    We set latent dim $d = 64$ and $K = 10$ here. The $p(\mathbf{o}|\mathbf{x}; \boldsymbol{\gamma})$, $q(\mathbf{x}|\mathbf{z}, \mathbf{o})$ are both parameterized by 5 deep stack of residual blocks [10], making the scale of images changing from $32 \times 32$ to $8 \times 8$ and reversely for $q(\mathbf{x}|\mathbf{z}, \mathbf{o})$. The network of $q(\mathbf{z}|\mathbf{o})$ is a simple 9-layer convolutional neural networks as adopted by recent deep semi-supervised methods [17]. The networks are trained for 200 epochs with Adam optimizer with learning rate $3 \times 10^{-3}$ and minibatch size $|B| = 100$ and $|S| = 200$.

**Crowdsourcing scheme**    For both MNIST and CIFAR-10, we random sample a subset of 4000 images from the training dataset and ask workers on Amazon Mechanical Turks (AMT) to annotate them. In each task, 36 images are sampled from the subset and shown to the worker. Each worker is asked to divide 36 images into any number of clusters. From these feedbacks, we can derive pairwise annotations for following experiments.

Figure 5: Comparison to deep MCVC on Face dataset with only a subset of 100 data points annotated. The experiment setting is the same as that in Fig. 3b.

## C    Addtional experimental results

**Comparison with deep MCVC [3]**    In Section 5.2, our model performs better than other competing methods because of the generative "instance model" and thus the usage of structure information in unlabeled data. Discriminative methods such as MCVC [3] cannot take advantage of the observations of those unannotated data, even with a more complex deep neural network. For fair comparison, we modify MCVC by using a more complex DNN (abbr. as deep MCVC). We test deep MCVC on the Face dataset, where deep MCVC uses the same network architecture (MLP with two hidden layers of 40 ReLU units) as our $q(\mathbf{z}|\mathbf{o})$ in our method. As shown in Fig. 5, the performance of deep MCVC is better than MCVC when more constraints are observed but worse than the proposed SCDC. The reason is that deep MCVC does not exploit the information of unannotated images during the training.

Furthermore, we also evaluate deep MCVC on CIFAR-10, whose best NMI score among 5 runs is 0.44. In contrast, the proposed SCDC achieves NMI score 0.55. The gap of the improvement is due to the help of unannotated data.