[Reviews · NeurIPS 2018]

Reviewer 1



A complex DGM is proposed that jointly models observations with crowdsourced annotations of whether or not two observations belong to the same cluster. This allows crowdsourcing non-expert annotations to help with clustering complex data. Importantly, the model is developed for the semi-supervised case, i.e., annotations are only observed for a small proportion of observation pairs. The authors propose a hierarchical VAE structure to model the observations, with a discrete latent-variable z \sim p(z | \pi), a continuous latent variable x \sim p(x|z), and observed data o \sim p(o|x). This is paired with a two-coin David-Skene model which is conditioned on the mixture variable z for annotations: L \sim p(L | z_i, z_j, \alpha, \beta), where \alpha and \beta are annotator-specific latent variables that model the "expertise" of the m_th annotator (precision and recall parameters, respectively). To the best of my understanding, through the dependence of the two-coin model on the latent mixture association, though it is not explicitly stated in the paper, z represents cluster association in the model. An amortized variational inference procedure is derived to train the model, with appropriate inference networks q(z, x | o) = q(z | o)q(x | z, o). Exact enumeration is used to integrate out z, and reparametrization sampling to integrate out x. The relational model is trained by maximizing the likelihood of the observed annotations conditioned on inferred z's. More interestingly, the authors leverage the work presented in Johnson et al. to derive a stochastic natural gradient variational inference procedure for the fully Bayesian version of the model. Remarkably, this results in a speed-up in training, as exact enumeration over the discrete latent variable is no longer required. Experimentation is carried examining the resulting clustering with a number of data-sets. The authors propose NMI as a quality metric for the learned clustering, and demonstrate that the proposed model consistently outperforms its competitors. Pros: - Solid modelling approach to the problem. The probabilistic model accounts for the important aspects of the application. - A fully Bayesian version of the model is proposed, and priors are chosen such as to improve model performance (e.g., sparsity inducing Dirichlet priors on the mixtures) - The authors go beyond standard amortized inference and derive the appropriate natural gradient AVI training procedure, leveraging the probabilistic structure of the model. - The model outperforms competitors in a range of experiments Cons: - The paper does not have a significant novel contribution, but rather composes and combines existing methods in a novel way for an interesting application. - The paper motivates the use of DGMs as they scale well to complex data. However, the usefulness of this is not demonstrated in the experimental section. The only "complex" dataset introduced is MNIST. The paper would be made significantly stronger if a real-world application with complex data was introduced, demonstrating the need for DGMs in this application. - The authors claim that BayesianSCDC is significantly faster to train than SCDC. This can intuitively be understood as exact enumeration is no longer necessary. However, BayesianSCDC may incur costs in dealing with other parameters (e.g., neural network parameters). It would be interesting to see a comparison of training times between the two versions of the model (this may be in the appendix if the authors cannot manage under the space constraints). Overall, I think this is a solid paper well suited to NIPS. The paper is clearly written and easy to follow. It builds on solid past work in the area of structered DGMs, and contributes a novel (to the best of my knowledge) probabilistic model to the crowdsourced clustering area. Specifically, enabling the use of partially-available annotations can make the use of crowdsourced annotations more feasible in a realistic setting. The quality of the work is high -- particularly the derivation of the natural gradient AVI procedure. However, the paper does not contribute novel modeling or inference tools, but rather combines existing work in a novel way. Further, I feel the usefulness of the model would be more clearly demonstrated if a real-world application with sufficiently complex data to require DGMs was experimented with.

Reviewer 2



Post Rebuttal Summary -- Thanks to the authors for a careful rebuttal that corrected several notation issues and provided an attempt at more realistic experiment using CIFAR-10 with real (not simulated) crowd labels. I found this rebuttal satisfactory and I am willing to vote for acceptance. I won't push *too* hard for acceptance because I still wish both the new experiment and the revised presentation of variational method details could go through more careful review. My chief remaining concerns are that it is still difficult to distinguish the reasons for gains from BayesSCDC over plain old SCDC because multiple things change (factorization of q(z,x), estimation of global parameters). I do hope the authors keep their promise to compare to a "middle of the road" 3rd version and also offer insight about how amortization makes the q(z|o)q(x|z,o) approach still less flexible than mean field methods. Review Summary -- Overall, I think the core ideas here are interesting, but I'm rating this paper as borderline accept because it doesn't do any experiments on real crowd-labeled datasets, misses opportunities to provide take-home insights about the various variational approximations attempted here, suffers from some minor bugs in the math (nothing showstopping), and leaves open several procedural questions about experiments. My final decision will depend on a satisfactory rebuttal. Paper Summary -- This paper considers the problem where a given observed dataset consists of low-level feature vectors for many examples as well as pair-wise annotations for some example pairs indicating if they should belong to the same cluster or not. The goal is to determine a hard/soft clustering of the provided examples. The annotations may be *noisy*, e.g. from a not-totally-reliable set of crowd workers. The paper assumes hierarchical model for both observed feature vectors O and observed pair-wise annotations L. Both rely on a shared K-component mixture model, which assigns each example into one of K clusters. Given this cluster assignment, the O and L are then generated by separate paths: observed vectors O come from a Gaussian-then-deep-Gaussian, and pairwise annotations L are produced by a two-coin Dawid-Skene model, originally proposed in [16]. Many entries of L may not be observed. Two methods for training the proposed model are developed: * "SCDC" estimates an approximate posterior for each example's cluster assignment z and latent vector x which conditions on observed features O -- q(z, x | O) -- but point estimates the "global" parameters (e.g. GMM parameters \pi, \mu, \Sigma) * "BayesSCDC" estimates an approximate posterior for all parameters that aren't neural net weights, using a natural-gradient formulation. Notably, for "SCDC" the posterior over z,x has conditional structure: q(z,x|o) = q(z|o) q(x|z,o). In contrast, the same joint posterior under the "BayesSCDC" model has mean-field structure with no conditioning on o: q(z,x) = q(z)q(x). Experiments focus on demonstrating several benefits: * 5.1 toy data experiments show benefits of including pairwise annotations in the model. * 5.2 compares on a Faces dataset the presented approach to several baselines that can also incorporate pairwise annotations to discover clusters * 5.3 compares the two presented methods (full BayesSCDC vs SCDC) on MNIST data Strengths -- * The pair-wise annotation model nicely captures individual worker reliability into a probabilistic model. * Experiments seem to explore a variety of reasonable baselines. Weaknesses -- * Although the method is intended for noise crowd-labeling, none of the experiments actually includes truly crowd-labeled annotations. Instead, all labels are simulated as if from the true two-coin model, so it is difficult to understand how the model might perform on data actually generated by human labelers. * The claimed difference between the fully-Bayesian inference of BayesSCDC and the point-estimation of global parameters in SCDC seems questionable to me... without showing the results of multiple independent runs and breaking down the differences more finely to separate q(z,x) issues from global parameter issues, its tough to be sure that this difference isn't due to poor random initialization, the q(z,x) difference, or other confounding issues. Originality -- The key novelty claimed in this paper seems to be the shared mixture model architecture used to explain both observed features (via a deep Gaussian model) and observed noisy pairwise annotations (via a principled hierarchical model from [16]). While I have not seen this exact modeling combination before, the components themselves are relatively well understood. The inference techniques used, while cutting edge, are used more or less in an "out-of-the-box" fashion by intelligently combining ideas from recent papers. For example, the recognition networks for non-conjugate potentials in BayesSCDC come from Johnson et al. NIPS 2016 [12], or the amortized inference approach to SCDC with marginalization over discrete local variables from Kingma, Mohamed, Rezende, and Welling [13]. Overall, I'm willing to rate this as just original enough for NIPS, because of the technical effort required to make all these work in harmony and the potential for applications. However, I felt like the paper had a chance to offer more compelling insight about why some approaches to variational methods work better than others, and that would have really made it feel more original. Significance -- The usage of noisy annotations to guide unsupervised modeling is of significant interest to many in the NIPS community, so I expect this paper will be reasonably well-received, at least by folks interested in clustering with side information. I think the biggest barriers to widespread understanding and adoption of this work would be the lack of real crowd-sourced data (all experiments use simulated noisy pairwise annotations) and helping readers understand exactly why the BayesSCDC approach is better than SCDC alone when so much changes between the two methods. Quality Issues -- ## Q1) Correctness issues in pair-wise likelihood in Eq. 2 In the pair-wise model definition in Sec. 2.2, a few things are unclear, potentially wrong: * The 1/2 exponent is just wrong as a poor post-hoc correction to the symmetry issue. It doesn't result in a valid distribution over L (e.g. that integrates to unity over the support of all binary matrices). A better correction in Eq. 2 would be to restrict the sum to those pairs (i,j) that satisify i < j (assuming no self edges allowed). * Are self-edges allowed? That is, is L_11 or L_22 a valid entry? The sum over pairs i,j in Eq. 2 suggests so, but I think logically self-edges should maybe be forbidden. ## Q2) Correctness issue in formula for mini-batch unbiased estimator of pair-wise likelihood In lines 120-122, given a minibatch of S annotations, the L_rel term is computed by reweighting a minibatch-specific sum by a scale factor N_a / |S|, so that the term has similar magnitude as the full dataset. However, the N_a term as given is incorrect. It should count the total number of non-null observations in L. Instead, as written it counts the total number of positive entries in L. ## Q3) Differences between SCDC and BayesSCDC are confusing, perhaps useful to breakdown more finely The presented two approximation approaches, SCDC and BayesSCDC, seem to differ on several axes, so any performance difference is hard to attribute to one change. First, SCDC assumes a more flexible q(x, z) distribution, while BayesSCDC assumes a mean-field q(x)q(z) with a recognition network for a surrogate potential for p(o|x). Second, SCDC treats the global GMM parameters as point estimates, while BayesSCDC infers a q(\mu, \Sigma) and q(\pi). I think these two axes should be explored independently. In particular, I suggest presenting 3 versions of the method: * the current "SCDC" method * the current "BayesSCDC" method * a version which does q(x)q(z) with a recognition network for a surrogate potential for p(o|x) (Eq. 10), but point estimates global parameters. The new 3rd version should enjoy the fast properties of BayesSCDC (each objective evaluation doesn't require marginalizing over all z values), but be more similar to SCDC. Clarity ------- The paper reads reasonably well. The biggest issue in clarity is that some some key hyperparameters required to reproduce experiments are just not provided (e.g. the Dirichlet concentration for prior on \pi, the Wishart hyperparameters, etc.). These absolutely need to be in a final version. Other reproducibility concerns: * what is the procedure for model initialization? * how many initializations of each method are allowed? how do you choose the best/worst? accuracy on training?

Reviewer 3



Summary =============== The authors tackle the problem of crowd-clustering in a semi-supervised setting. They use a deep generative model, more specifically a density network, to model the instances’ features. This is combined with a latent variable model to represent the noise in the labels provided by the different annotators. The propose a structural variational inference algorithm based on natural gradient descend to infer the expertise of the annotators, the ground truth (for both, the instances of the training with and without annotations) and the parameters of the deep generative model that is used to predict the ground truth of new unobserved instances. The novelty of the paper lies in the “instance model” (the relation between the ground truth clustering and the observations) but relying in state-of-the-art deep generative model. However, there is no novelty in the “annotation model” (relation between the ground clustering and the annotations). The experiments need more details to evaluate the fairness of the comparison against the different baselines they propose. Details =============== The model proposed in the paper can be divided in the “annotation model” (generative model of the annotations given the latent ground truth clustering) and the “instance model” (generative model relating the ground truth clustering with the features of the instances). 1) Annotation model: The authors adapt the method proposed by David and Skene for classification to the problem of clustering: each instance is associated with a latent variable representing the ground truth label (cluster the instance belongs to). Based on this unobserved ground truth partition and the latent expertise of the annotators (sensitivity and specificity) the annotations are generated following a Bernoulli distributions. This model is not novel: it has been previously been proposed in the literature [3] Multiple clustering views from multiple uncertain experts (ref. in paper). 2) Instance model: Here is where the novelty of the paper lies (from a modeling perspective). Instead of using a discriminative model (p(x|o)) they use a generative model (p(o|x)). This allows the authors to take advantage of unlabeled instances. The paper is clear and well-written however the details given in the experimental section are not enough to understand why the method is outperforming the baselines. a) The authors do not give any details about the deep generative model they use for modeling p(o|x) (Experiments are not reproducible with the data provided in the paper. Although the authors claim that the code will be publish, these details should also appear in the paper or in the supplementary material). Secondly, it is not clear to me whether the method is performing better because of the generative nature of the “instance model” or just because this model is just most powerful. What would be the outcome of [3] if it is trained with a state of the art DL network? Given z (or an estimation of z) the problem is a standard semi-supervised learning algorithm and a better semi-supervised model will perform better. b) The annotations are artificially simulated: The simulation is a simplification of reality that does not take into account several effects (correlations between the annotators, homogeneous expertise of the annotators across the instance space, not-uniform distribution of annotations across instances/annotators …). From the inference perspective, the authors resort to two different methods: Amortized structured variational inference for the SCDC method (similar to the semi-supervised variational auto encoder) and a natural gradient with a recognition network for the BayesSCDC that it is nicely motivated by the fact that it avoids the linear scaling with the number of clusters. More details about the inference and the recognition network should be provided, either in the paper or in the supplementary material.